# Turning up the heat mimics allosteric signaling in imidazole-glycerol phosphate synthase

Federica Maschietto[1,8] ✉, Uriel N. Morzan[2,8] ✉, Florentina Tofoleanu[1,3,7,8], Aria Gheeraert [4,5,8], Apala Chaudhuri[6,8], Gregory W. Kyro [1], Peter Nekrasov[1], Bernard Brooks[3], J. Patrick Loria [1,6] ✉, Ivan Rivalta [4,5] ✉ & Victor S. Batista [1] ✉

Allosteric drugs have the potential to revolutionize biomedicine due to their enhanced selectivity and protection against overdosage. However, we need to better understand allosteric mechanisms in order to fully harness their potential in drug discovery. In this study, molecular dynamics simulations and nuclear magnetic resonance spectroscopy are used to investigate how increases in temperature affect allostery in imidazole glycerol phosphate synthase. Results demonstrate that temperature increase triggers a cascade of local amino acid-to-amino acid dynamics that remarkably resembles the allosteric activation that takes place upon effector binding. The differences in the allosteric response elicited by temperature increase as opposed to effector binding are conditional to the alterations of collective motions induced by either mode of activation. This work provides an atomistic picture of temperature-dependent allostery, which could be harnessed to more precisely control enzyme function.

Allosteric mechanisms are ubiquitous in biological macromolecules and are essential to transmit the perturbational effect of ligand binding at one site to another, often distant, functional site[1–4]. Drugs that target allosteric mechanisms offer significant advantages over traditional orthosteric modulators, including enhanced selectivity in tuning responses and intrinsic safeguards from overdosage[1,5–11]. However, the lack of molecular-level understanding of the nature of allosteric mechanisms often remains a major impediment for design and development of allosteric drugs[4,12–14]. It is thus critical to establish paradigms for regulatory processes in prototypical allosteric enzymes[15–22] to gain atomistic insight into the driving forces of

allosteric mechanisms, expand the scope of enzyme engineering, and open new avenues for drug discovery[23–28].

Temperature plays a crucial role in allosteric mechanisms[29–32], especially in thermophilic enzymes that remain active at elevated temperatures. The elevation of temperature has been linked in some K-type enzymes[30] to an inversion of allosteric modulation and in others to an enhancement of modulation[32,33]. In all these cases, the temperature-induced effects are a consequence of an adjustment in substrate affinity upon allosteric ligand binding. Other studies have correlated temperature elevation with enhanced enzymatic activity. However, these studies did not elucidate the role of the temperature in

[1]Department of Chemistry, Yale University, P.O. Box 208107 New Haven, CT 06520-8107, USA. [2]International Center for Theoretical Physics, Strada Costiera 11, 34151 Trieste, Italy. [3]Laboratory of Computational Biology, National Heart, Lung and Blood Institute, National Institutes of Health, Bethesda, MD 20852, USA. [4]ENSL, CNRS, Laboratoire de Chimie UMR 5182, 46 allée d'Italie, 69364 Lyon, France. [5]Dipartimento di Chimica Industriale "Toso Montanari", Alma Mater Studiorum, Università di Bologna, Bologna, Italy. [6]Department of Molecular Biophysics and Biochemistry, Yale University, New Haven, CT 06520, USA. [7]Present address: Treeline Biosciences, 500 Arsenal Street, Watertown, MA 02472, USA. [8]These authors contributed equally: Federica Maschietto, Uriel N. Morzan, Florentina Tofoleanu, Aria Gheeraert, Apala Chaudhuri. ✉e-mail: federica.maschietto@yale.edu; umorzan@ictp.it; patrick.loria@yale.edu; i.rivalta@unibo.it; victor.batista@yale.edu

the allosteric activation[34,35]. An understanding of the physico-chemical features that underlie this phenomenon could have profound implications for allosteric drug design against proteins from extremophilic organisms or for enzyme regulation under harsh conditions[36,37].

Here, we explore the temperature-dependent allostery of imidazole glycerol phosphate synthase (IGPS) from the thermophile *Thermotoga maritima* (*T. maritima*), shown in Fig. 1[29,38]. This V-type allosteric enzyme, composed of a glutamine amidotransferase (HisH) and cyclase (HisF) subunit, is absent in mammalian organisms, rendering it a potential therapeutic target for various pathogens. Moreover, since it plays a central role in metabolism, deletion of the IGPS gene in bacteria results in increased sensitivity to antibiotics[39], and a decrease in infectivity[40]. The allosteric mechanism of this prototypical enzyme at room temperature has been extensively investigated[41–48], proving the reliability of computational predictions in detecting the fundamental allosteric signaling mechanism from the early IGPS dynamics upon effector binding[47,49–51].

The allosteric ligand N′-[(5′-phosphoribulosyl)formimino]−5-aminoimidazole-4-carboxamide-ribonucleotide (PRFAR) features an entropically-driven binding to the HisF subunit at room temperature, enhancing glutamine hydrolysis 5000-fold over its basal level. Similarly, increasing the temperature of the native *T. maritima* environment drastically enhances millisecond dynamics in both PRFAR-free (apo) and PRFAR-bound (holo) IGPS[29]. Furthermore, the catalytic enhancement in the holo IGPS is nearly independent of temperature in the 303–350 K range[29]. In contrast, basal levels of Gln hydrolysis increase sharply from 303 to 350 K resulting in PRFAR being a weaker activator at the physiological temperature for *T. maritima* growth[29]. It has also been suggested that the dynamics of the apo enzyme become comparable to the PRFAR-bound form at 50 °C, whereas at 30 °C there

is a substantial difference between these two states[29]. These data suggest that increasing the sample temperature produces a similar effect on protein dynamics and allosteric activation as the binding of the natural allosteric ligand, PRFAR.

Here, we elucidate the effect of temperature on the allosteric mechanism at the atomistic level. We show that both higher temperatures and binding of the allosteric ligand PRFAR increase flexibility in some regions outside of the effector site in the heterodimeric enzyme IGPS. This allows for conformational sampling of an active enzyme form. We also find that motions induced by the orthosteric effector propagate through well-defined secondary structure elements and chemical interactions between specific residues that are analogous to those caused by the temperature increase. The collective motions that promote the activation differ in the two cases, supporting the hypothesis that allosteric signaling is highly adaptable and can be functionally compensated for by means beyond mutations[50,52] or small molecule activators[43].

## Results and discussion

### Effector binding vs. temperature increase: dynamical aspects

The allosteric activation of IGPS by PRFAR is weaker at higher temperatures[29]. This is because the Arrhenius dependence of the catalytic activity is steeper in the basal state than in the PRFAR-activated IGPS. At higher temperature PRFAR is less activating as an allosteric ligand enhancing activity by two orders of magnitude vs. three at room temperature[29]. We inspect correlated pathways from Cα displacements simulated at 30 °C and 50 °C in the apo and holo states through the Eigenvector centrality (EC) metric. The EC entries are scores indicating how central each residue is to the intercommunication established by nuclear motions, pinpointing key amino acids that participate in the IGPS dynamics[44]. Our results show that the difference in the EC distribution of two equilibrium states enables the recognition of the main features associated with the transition between these two states, providing unique insights into the allosteric signaling process.

Figure 2a shows the EC difference associated with PRFAR binding at 30 °C (left panel), 50 °C (right panel) and with the 30 °C → 50 °C temperature increase in the apo-IGPS (middle panel). The change in the EC indicates how the increase in temperature or the binding of PRFAR influences the relative contribution of each residue to the dynamics of IGPS. Remarkably, the EC trends upon temperature increase in apo-IGPS show a strong similarity to those observed subsequent to PRFAR binding at 30 °C.

For the sake of clarity, IGPS can be divided in two sides, as illustrated in Table SI1 and Fig. 2, i.e., *sideR* and *sideL*, since a signature for the allosteric activation is the large increase in EC on loop 1 (HisF), *f*α2 (HisH), and *h*α1 (HisH) at *sideR* along with a depletion of EC at *sideL*, as previously observed[44,46,53]. In contrast, the changes in EC associated with the PRFAR binding process started at 50 °C are much more homogeneous amongst the residues, and qualitatively different from the behavior at 30 °C. This suggests that both activators elicit an allosteric response, although of different magnitude. Besides, the presence of the effector has a substantially different effect on the protein dynamics at 50 °C, which can be connected to the much weaker PRFAR-induced activation at higher temperatures[29]. This behavior is also in agreement with the observed increased mobility of PRFAR in the *holo50* simulations, leading to significant displacements in the pocket that were absent in the same timescales in the room temperature dynamics (Supplementary Fig. 1).

In order to understand how the crosstalk between the effector binding site and the active site of IGPS is modified by temperature, we studied the optimal residue-to-residue communication channels connecting the PRFAR phosphate binding sites with the catalytic site in HisH. Figure 2b shows the amino acids belonging to these channels, distinguishing between "external" (i.e., solvent exposed) and "internal" (i.e., buried in the protein matrix) pathways. The *apo30* has mostly

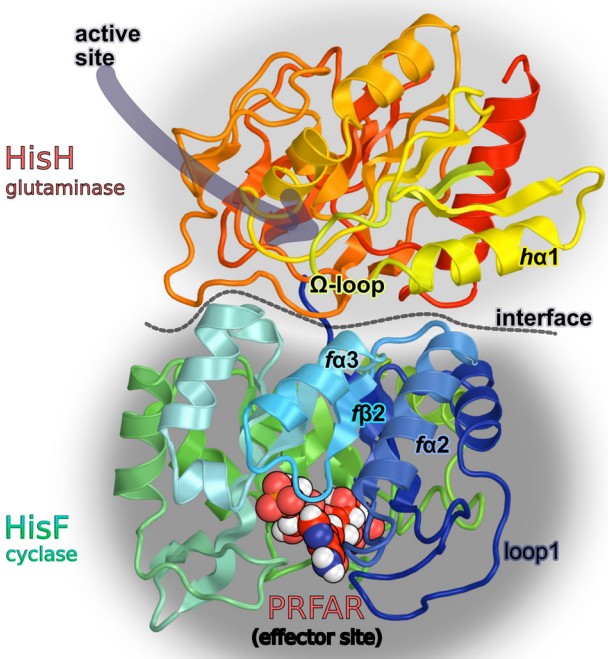

**Fig. 1 | Molecular representation of IGPS.** IGPS is a bienzyme composed of two subunits, i.e., HisF and HisH, that constitute the cyclase and glutaminase domains, here colored respectively in green-to-blue and red-to-yellow gradients, respectively, and separated by a dotted line which marks the interface between HisF and HisH. The labels (*f*α2, *f*α3, *f*β2, loop1, *h*α1, Ω-loop) indicate secondary structure elements that are directly involved in the allosteric regulation, as found by previous studies.

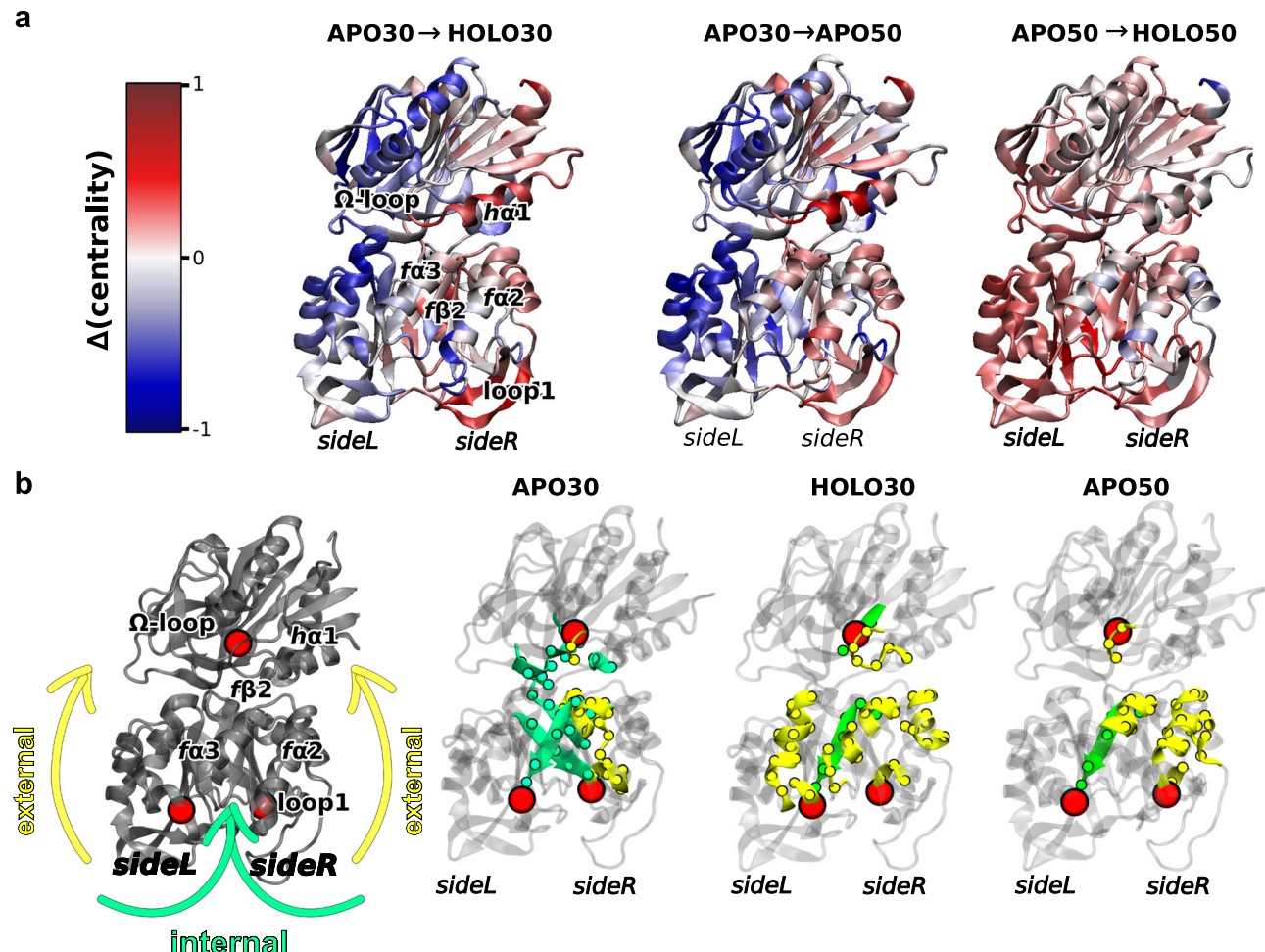

**Fig. 2 | Comparison of allosteric signals in temperature- and effector-induced modulations. a** Eigenvector centrality difference associated with binding of PRFAR effector (holo enzyme) at 30 °C (left) or 50 °C (right) and with a temperature increase in the apo IGPS enzyme (middle). Secondary structures with increased (in red) or decreased (in blue) centralities are analogous upon effector binding or temperature increase in both HisF and HisH units of IGPS. Secondary structures mainly involved in the temperature- and effector-induced allosteric communication network (positive differential centralities, in red) are located at *sideR*, including loop1, *f*α2, *f*α3 in HisF and *h*α1 in HisH. The effector binding at high temperature (50 °C) features much fewer specific communication pathways than at 30 °C.

**b** Comparison of specific mutual information pathways connecting the PRFAR phosphate binding sites (residues *f*T104 and *f*A224 in HisF, red spheres located in *sideL* and *sideR*) to the active site (*h*G50 in HisH, red sphere). Secondary structure elements involved in the shortest communication pathways are highlighted, separating contributions of surface (solvent-exposed) amino acids ("external", in yellow) from those buried in the protein ("internal", in green). The information pathways in the allosterically inactive apo enzyme at 30 °C (*apo30*) differ from those of the IGPS activated by a temperature increase (*apo50*) or by effector binding (*holo30*), which instead feature some similarities. Source data are provided as a Source Data file.

internal communication pathways, while the *holo30* and *apo50* have a significantly higher proportion of external residues. This internal-to-external transition suggests that temperature increase and effector binding have a parallel effect on the signaling pathways.

The effect of temperature increase can be regarded as an alternative route to activate the fluctuation of external amino acids, increasing the influence on the helices *h*α1 (HisH), *f*α1, *f*α2 (HisF) and the omega loop involved in the allosteric activation. This activation shows clear similarities between the effector binding effect and the temperature increase (Fig. 2b). In both cases, we observe a strong internal-to-external transition in the communication pathways going from the *apo30* system to the active ones (*apo50* and *holo30*), which suggests that this transition might be a key factor in determining the catalytic activation and IGPS thermostability.

The interdomain hinge-like (breathing) motion has been recognized as one of the important elements of the IGPS allosteric signaling mechanism at room temperature[29,46,47]. PRFAR binding, in fact, slightly reduces the breathing motion angle (as defined by the Cα of residues *h*G52, *h*W123, *f*F120, and *h*E96,

*h*W123, *f*F166 (Supplementary Fig. 2), while significantly shrinking the distribution of angle amplitudes explored by the IGPS complex. Moreover, as previously shown for 100 ns MD simulations[44,46,51], these large angle fluctuations in the *apo30* simulation have slower oscillation frequencies with respect to those in *holo30*. The breathing motion in *apo50*, instead, features an angle average comparable to *holo30* and a much broader *h*G52, *h*W123, *f*F120 angle distribution (see Supplementary Fig. 2). These results suggest that an increase in temperature has a similar effect on the local dynamics of allosteric propagation as binding of the effector. However, the temperature increase seemingly has less control over allosteric collective motions. This hypothesis fits the notion that allostery is not a binary process, resulting in "partial allosteric effects" as reported for other systems[54]. To ensure reproducibility and statistical relevance of our outcomes, we analyzed correlations and fluctuations for three additional replicas for each state (apo30, holo30 and apo50), shown in Supplementary Figs. 3–5. The high correspondence of the additional replicas to the set of simulations presented in the

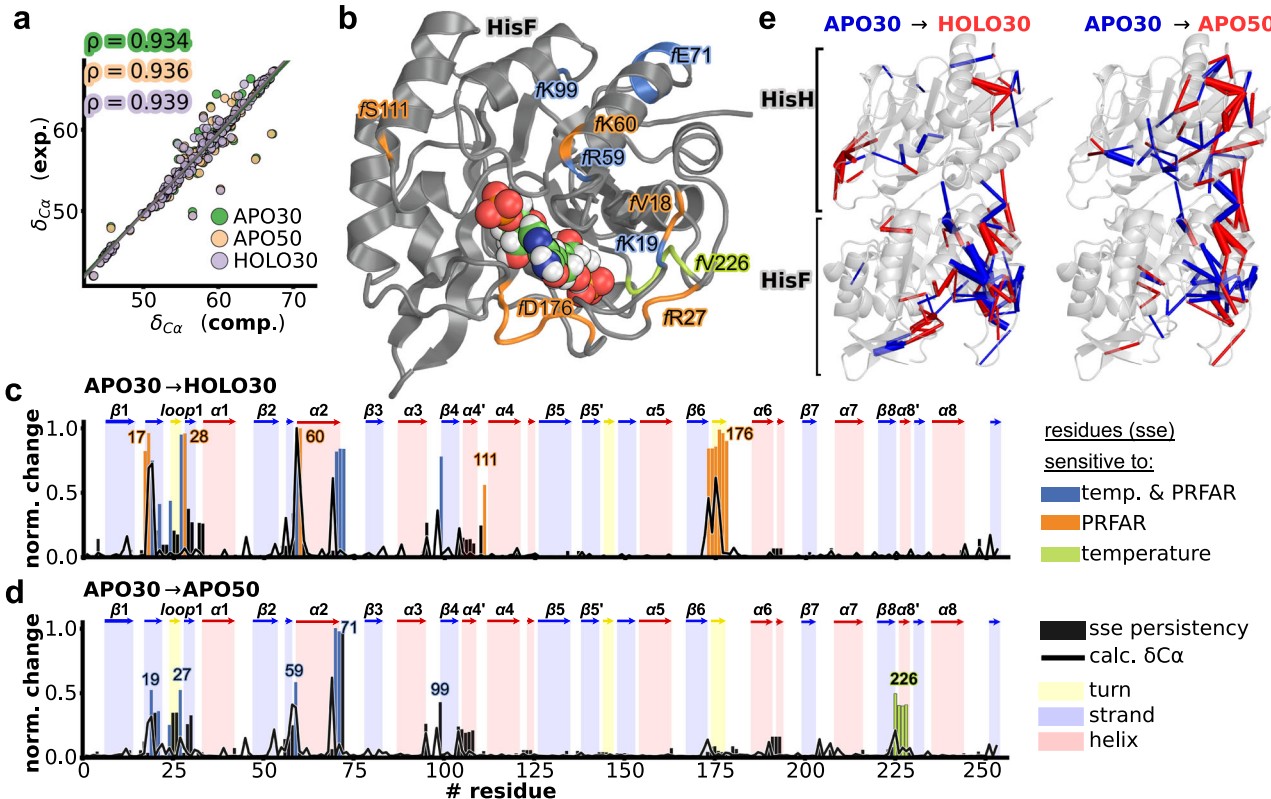

**Fig. 3 | Correspondence between dynamics from the MD trajectories and secondary structure changes. a** Correlation between computed average Cα chemical shifts obtained by employing SHIFTX2[92] computations over *apo30* (green), *apo50* (purple) and *holo30* (pale orange) MD simulation runs and corresponding values obtained experimentally. Pearson correlation values are shown on the top left outlined with matching colors. **b** Cartoon representation of HisF, with selected residues colored according to (**c**) and (**d**). **c** Change in average SHIFTX2[92] simulated Cα chemical shift (solid black line) calculated from *apo30* to *holo30*. Bars represent the change in secondary structure element (SSE) persistency for each residue. SSE persistency values are computed as the percentage of frames in which each residue retains the secondary structure as assigned in the 1GPW crystal structure. **d** Analogous to (**c**) but depicting the SSE and Cα changes occurring from *apo30* to *apo50*. Overall, residues whose secondary structure changes as a result of the

presence of PRFAR are colored in orange, while residues in the same region that are equally affected by temperature and PRFAR are colored in marine blue. Residues colored in lime green display significant secondary structure changes only in the *apo30* to *apo50* profile but are unperturbed by PRFAR. Topography of secondary structure elements as assigned in the 1GPW crystal structure is shown in background in (**c**, **d**) and delimited by arrows: red, blue, yellow regions correspond to residues organized in helical, (β)-strand and (β)-turn structures, respectively. **e** Perturbation contact networks between *apo30* and *holo30* (left panel) and between *apo30* and *apo50* (right panel), showing the most relevant contact perturbations (i.e., a weight threshold of six contacts). Blue and red edges represent decrease and increase, respectively, of contacts upon effector binding (left panel) or temperature increase (right panel). Edge widths are proportional to the differences in number of atomic contacts. Source data are provided as a Source Data file.

main text is confirmed by small divergence in the RMSD profiles, with correlation of RMSF distributions for the three new replicas all exceeding 0.9.

**Allosteric activation vs. temperature increase: a structural perspective**

To verify our hypothesis, we analyze and compare the structural changes associated with allosteric activation and temperature increase (see Fig. 3, showing a selected set of simulations and Supplementary Fig. 9, showing the average results of four independent simulations), with the aim of identifying the most relevant secondary structure changes triggered by either mode of activation. NMR is highly sensitive to subtle changes in protein structure and is extremely powerful for quantifying how the dynamic equilibrium is perturbed by an effector or by the temperature. We performed NMR backbone chemical shift measurements on HisF, which is the IGPS protein with the largest number of signal assignments and that hosts the effector binding site. Both activators induce signal broadening resulting from large fluctuations near the effector site as well as the dimer interface. Hence, we compared experimental shifts with those computed from our MD simulations employing the SHIFTX2 package[55]. We estimate NMR shifts for each state by averaging over 10,000 configurations extracted from

the corresponding 1μs MD simulation (replica 0). Using computed values allows us to account for those residues that are hard to assign experimentally, which is the case at elevated temperature conditions or for exchange-broadening upon effector binding. The full list of NMR assignments is provided as supplementary material. For experimental ΔCα chemical shifts profiles, we refer to the SI (Supplementary Fig. 6). Regardless of the state under consideration (*apo30, holo30* or *apo50*), computed and experimental values show near-perfect agreement (Fig. 3a), with Pearson correlation coefficients above 0.9. This observation suggests that the structural ensemble derived from MD is highly correlated to that of NMR at the molecular level, substantiating our analysis.

Figure 3b, c shows the overlay of simulated difference in Cα shifts and secondary structure changes that take place during the transition from *apo30 to holo30* and from *apo30 to apo50*. ΔCα values are computed as the normalized difference between the two sets (using the same basis for the normalization). Secondary structure changes are computed as the difference in secondary structure (SSE) persistency between the two states. SSE difference persistency values close to one are indicative of large perturbations elicited by the activator (temperature or effector). Clearly, dynamical changes in secondary structure show a high degree of correlation with average ΔCα simulated

shifts supporting the notion that our computational predictions are well aligned with experimental data also at high temperature, in analogy with previous studies at room temperature[47,49–51].

In agreement with the dynamical changes discussed in the previous section, there are important similarities between the structural rearrangements associated with the 30 °C-to-50 °C temperature increase and those associated with PRFAR binding at 30 °C. One of the main aspects of this resemblance is the conformation of loop1 in HisF (residues $f$V17-$f$D31). In *apo30*, it adopts a combined β-sheet/helix structure but is mostly devoid of regular secondary structure in both *apo50* and *holo30*. This secondary structure change can be associated with the increase in flexibility of loop1, which has been suggested to play an important role in the activation process[50]. Furthermore, helix $f$α2 ($f$R59-$f$E71), which has been identified as being involved in the allosteric pathway of IGPS[44,46,51], shows an almost identical structural response to temperature increase and to PRFAR binding. In particular, residues that are equally perturbed by the effector and temperature increase are colored in marine blue in Fig. 3. Residues that experience large changes in response to PRFAR are shown in orange. Interestingly, both temperature- and effector-induced perturbations are largely localized in loop1 and $f$α2 regions, suggesting that the temperature increase mimics to a large extent the perturbations caused by PRFAR.

This perturbation of secondary structure elements extends to $f$α1 (residues $f$R59, $f$K69, $f$E71) and $f$α2/$f$β2 (residues $f$K99 and, to a lesser extent, $f$R95), which are located in a critical spot for the allosteric transmission, and are present in all the optimal signaling pathways presented in Fig. 2, supporting the observation that both temperature and the effector elicit an allosteric response and signal transmission to the top of the cyclase domain at the interface with HisH. They also belong to the group of amino acids with higher EC increase upon PRFAR binding or temperature increase (see Fig. 2a). Moreover, residue $f$R95 has been previously identified as one of the key locations in the allosteric transmission from the HisF to the HisH domain[51] and residue $f$K99 belongs to the ammonium ion gate of the HisF barrel, which is hypothesized to open upon effector or substrate binding[48,56].

Aside from these similarities, temperature increase and PRFAR binding lead to only few different outcomes, some of which, as could be expected, in the region near the effector site. For the selected simulations compared in Fig. 3 (replica 0), an increased helicity in $f$β6-$f$α6 is observed in holo30 but almost absent in apo50. The formation of these helices is mostly triggered by the interactions with PRFAR (see Supplementary Fig. 8). When considering additional replicas in the analysis, the average effect of temperature and PRFAR becomes even closer (Supplementary Fig. 9), supporting our hypothesis that the effect of temperature closely mimics that of PRFAR. The RMSF difference plots (Supplementary Fig. 7) also show an increase in stiffness in holo30 in the $f$β6-$f$α6 and $f$β7-$f$α7 turns near the effector site that is only barely present in apo50. The $f$β8-$f$α8 region (lime green in Fig. 3 and Supplementary Fig. 9), instead, is slightly more affected by temperature than by the presence of PRFAR. Only a few residues are affected exclusively by a temperature increase when the average over all replicas was considered (Supplementary Fig. 9), including G15, T21, A54, V107, N109, A117, Q118, K162, L215, L222, K242.

To further characterize the parallel between temperature increase and effector binding, we performed the DPCN analysis (Fig. 3e) to compare the changes in contacts upon PRFAR binding (left panel) with those found in the apo protein when the temperature increases from 30 °C to 50 °C. Overall, the contact perturbation networks induced by PRFAR binding and temperature increase (from 30 °C to 50 °C) are highly correlated. The computed Spearman correlation coefficient between residue pairs carrying a contact weight greater or equal than 6 is over >0.9 in all sets, clearly indicating the presence of similarities between the two activations of apo IGPS with notable deviations in the respective difference networks (Supplementary Fig. 10).

In both cases, the majority of contact alterations are located at the *sideR* of the protein (Fig. 3e), in analogy with the eigenvector centrality analysis (see Fig. 2a). The DPCN results are also consistent with the signaling pathways analysis (see Fig. 2b). Most of the contact perturbations due to both PRFAR binding and temperature increase involve solvent-exposed residues at the protein surface.

The detected contacts involve essentially the same set of nodes and edges (including sign of the perturbation), the differences being mostly in the absolute numbers of contact changes (i.e., the "perturbation intensities"), in agreement with what is observed in Fig. 2c, d and Supplementary Fig. 7. In particular, the alterations of the salt bridge network between $f$α2, $f$α3 and $h$α1 that have been recognized for the allosteric pathway of holo IGPS[44,46,51] also appear upon temperature increase. Moreover, the number of contacts between the $f$α2 and $f$α3 helices increases in an almost identical way in the two cases, while the loss of contacts in $f$α2 and $f$α3 is elicited by PRFAR binding more than by the increase in temperature. In contrast, all changes in contacts between $f$α2 and $h$α1 are more pronounced with temperature increase than with effector binding.

To further characterize the differences near the effector site, we looked more closely at the perturbations induced around specific nodes belonging to the $f$α7-$f$β7 and the $f$β6-$f$α6 turns near the effector binding site (what are called the "induced perturbation network", IPN). For instance, the presence of PRFAR increases contacts between residues $f$G202 and $f$G203 (directly in contact with PRFAR) and residues $f$R175, $f$S172 and $f$K179 as shown by the IPN of residue $f$D176 in the $f$β6-$f$α6 turn (Supplementary Fig. 11).

The most recent PDB structure, 7AC8[47] shows an IGPS mutant in its active conformation. In this conformation, residue $f$D176 forms a salt bridge with $f$K19, which in turn forms a salt bridge with the PRFAR glycerol phosphate group. These changes indicate a propagation of contact perturbations that is consistent with our analysis of the secondary structure changes in the $f$β6-$f$α6 region (see Fig. 2c) and is supported by the RMSF difference profiles (Supplementary Fig. 7). Temperature increase also causes a gain of molecular contacts in the $f$β8-$f$α8 turn, but not to the same extent as for PRFAR binding. This is in line with the SSE perturbation in this region with rising temperature (see Fig. 2d and Supplementary Fig. 7).

The mechanism of Glutamine hydrolysis in the HisH active site involves reorganization of an oxyanion hole[57,58], to facilitate nucleophilic attack of the cysteine residue ($h$C84) binding the Gln substrate (e.g., see Fig. 9 in ref. 46). This reorganization in the oxyanion hole region involves nearly 180° rotation of hG50 (of the conserved 49-PGVG sequence) to place a stabilizing HB between the $h$V51 residue (nearby in sequence) and the Gln substrate, which is possible only upon breaking of the backbone HB existing between $h$V51(N) and the $h$P10(O) residue lying in the W-loop (see Fig. 1). Thus, a strong $h$V51(N)-$h$P10(O) HB is typical of the apo state, as it holds the 49-PGVG sequence in an inactive conformation, inapt for catalysis. It was experimentally shown that PRFAR binding causes significant motions in this region of the HisH enzyme[57] and MD simulations showed that such motions are, indeed, the consequence of breaking of the $h$V51(N)-$h$P10(O) HB and flippling[46].

Here, to verify if the temperature increase has effects in the HisH active site similar to those observed for PRFAR binding, we report experimental evidence of chemical shifts alterations induced by temperature (see Supplementary Fig. 12, showing that $h$G50 resonance move from slow to intermediate, to fast exchange upon temperature increase from 20 °C to 50 °C) and the time evolution of the critical $h$V51(N)-$h$P10(O) HB across the four 1μs MD simulation replicas (see Supplementary Fig. 13). We observed that this critical HB holding the structure in the inactive conformation is essentially broken in apo50, much closer to what found in holo30 (where is often broken but it can be reformed) than to apo30, where is found to be strong, as expected, and only rarely it breaks. These results are, thus, consistent with the

presence of a basal activity of IGPS in absence of effector at room temperature (apo30) and they further confirm that the temperature increase to 50 °C could affect the catalytic activity by altering the local dynamics in the active site in the same fashion as the effector binding.

In summary, *apo*50 and *holo*30 differ mostly in the way the local regions near the binding site for effectors are structured. This suggests that when an effector binds, it not only causes specific allosteric changes (which spread to the active site), but also other local changes that the protein absorbs. These results emphasize the differences in contacts at the PRFAR binding site caused by an effector binding, *versus* a temperature increase and how these are connected to the overall structure changes.

## Molecular-level description of NMR chemical shifts and temperature-dependent dynamics

In the previous section, we showed the qualitative agreement between the simulated and experimental NMR predictions. Here, in order to reach a molecular-level description of the dynamical changes induced by effector and temperature increase, we analyzed the dynamics of specific residues involved in chemical shift changes. Rather than Cα shifts we now consider $^1$H shifts, which are directly correlated with changes in HB profiles thereby supporting a molecular-level interpretation associated with bond-forming/breaking or strengthening/weakening events. The correlation between $^1$H experimental and simulated shifts is shown in Supplementary Fig. 4.

Of the 253 residues in the HisF subunit, we shortlisted 164 residues, which were non-overlapping, over the temperature range, in the NMR HSQC spectra, and which showed an unambiguous temperature shift across the temperature range of 20–50 °C. For these selected residues, we determined the temperature coefficient both in the neighborhood of 30 °C (between $T_1 = 292.92$ K and $T_2 = 302.73$ K) and 50 °C (between $T_1 = 307.62$ K and $T_2 = 322.41$ K) as $\partial(\partial_{HN})/\partial T = (\partial_{HN,T2} - \partial_{HN,T1})/(T_2 - T_1)$. Non-linear changes in proton chemical shift over temperature has been suggested to be an indicator of conformational change for the given residue[59], thereby providing a suitable starting point to investigate temperature dependent modification of the environment for those amide protons around 30 °C and 50 °C.

Thirty-two residues display "curvature" in the slope of $^1$H chemical shift changes over temperature. The rest of the residues display the characteristic linear shift with the increase in temperature that is typically observed in proteins[60]. To discern temperature-dependent chemical shift curvature in residues quantitatively, we conducted an extra-sum-of-squares F-test to compare linear versus quadratic fits, with the linear model as the null hypothesis, and those residues with a $P$ value of <0.05 were classified as residues displaying "curvature".

Figure 4 shows the temperature-dependent evolution of the five residues with the most prominent changes in NMR temperature coefficient, i.e., those that feature a significantly different temperature coefficient around 30 °C with respect to that around 50 °C. Among those top 5 residues, residue *f*L63 is the only one displaying a positive

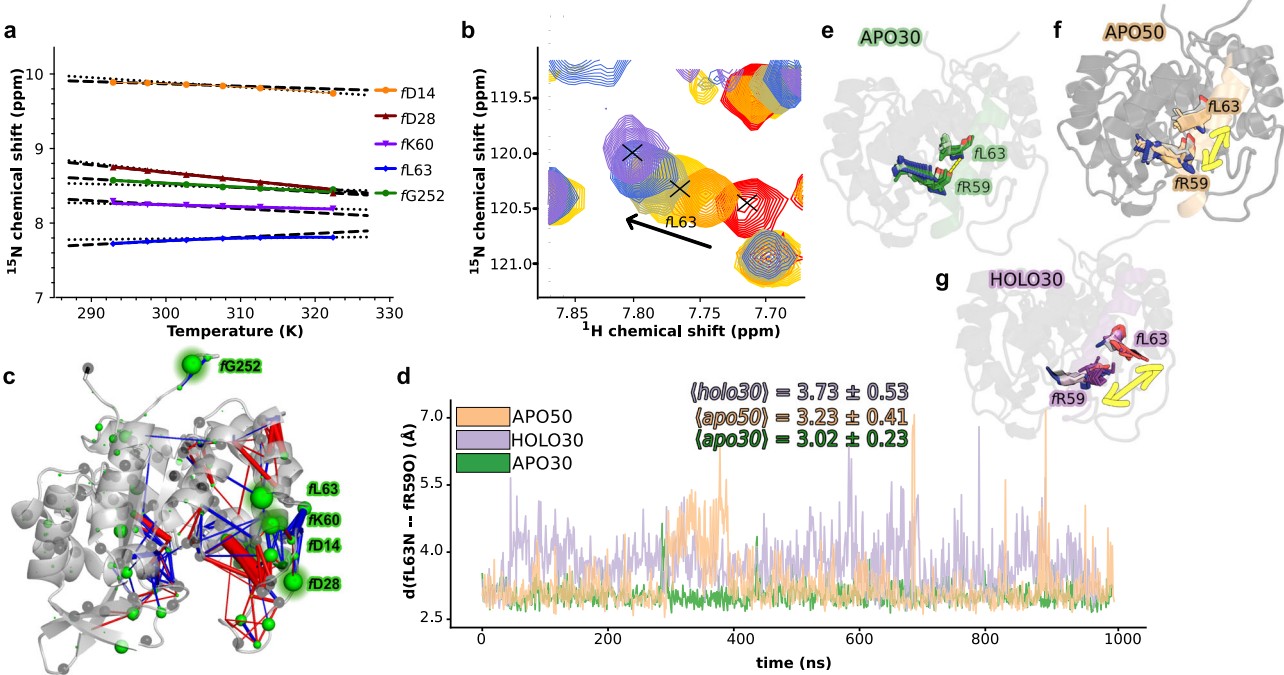

**Fig. 4 | Temperature-dependent NMR chemical shifts and temperature-dependent molecular dynamics. a** Experimental chemical shifts (relative to the APO state) of the five residues with the biggest change in temperature coefficient around 30 °C (between 292.9 K and 302.7 K, with slope displayed by dashed line) and around 50 °C (between 307.6 K and 322.4 K, with slope displayed by dotted line). Typical experimental error bars for the temperature coefficients are too small to be visualized (<1 ppb). **b** Experimental spectral overlay for the most prominent change in NMR chemical shifts (i.e, NH of *f*L63) at all temperatures under consideration. **c** Superimposition of the experimental temperature coefficient changes between 30 °C and 50 °C and the asymmetric dynamical perturbation contact network between all backbone NH moieties and the rest of the protein. The absolute variation between temperature coefficient around 30 °C and 50 °C are displayed in green spheres centered on the nitrogen atoms of N-H groups, with sphere sizes being proportional to the variation in the slope (ppb/K). Gray spheres refer to residues that could not be assigned unambiguously over the full temperature range in HisF, thus missing temperature coefficient values. In the perturbation network, blue and red edges represent a decrease and increase, respectively, of contacts upon an increase in temperature. Edge widths are proportional to the differences in number of contacts. **d** Evolution of the *f*L63-*f*R59 backbone hydrogen bond (defined as the distance between the backbone atoms *f*L63-N and *f*R59-O) over time in the apo30, apo50, and holo30. **e–g** Overlay of 100 configurations sampled (1 every ten ns) for the *f*L63 and *f*R59 residues along the apo30, holo30 and apo50 MD trajectories. The mean and standard deviation of the *f*L63-*f*R59 backbone hydrogen bond distance are also indicated. **e–g** Overlay of 100 *f*L63L and *f*R59 configurations sampled (1 every ten ns) along the *apo30*, *holo30*, and *apo50* MD trajectories. While in *apo30* *f*L63 and *f*R59 form a tight hydrogen bond, which is almost never present in *holo30* (as highlighted by the long yellow arrow in **g**). *apo50* represents an intermediate state between the two extremes, where the *f*L63 and *f*R59 become less stable upon temperature increase. Source data are provided as a Source Data file.

temperature coefficient. Upon the temperature increase, the positive slope diminishes significantly around 307 K and thereafter is near constant. Residues $f$G252 and $f$K60, featuring negative temperature coefficients, also present a change in slope at increasing temperature that tends to alleviate the temperature dependence (i.e., the slope becomes less negative around 50 °C than around 30 °C). On the contrary, residues $f$D14 and $f$D28, while featuring negative temperature coefficients, showed more negative slope around 50 °C than around 30 °C. As shown in Fig. 4c, all of these top five residues are located at *sideR* near the allosteric pathways, with the notable exception of residue $f$G252 located at the C-term loop of HisF. In addition, these also fall within hydrogen bonding distance to the regions that showed the largest secondary structural changes, such as ($f$β1, loop1) and $f$α2 (Fig. 2c, d).

Interestingly, the observed temperature coefficient changes from experimental shifts correlate nicely with the asymmetric dynamical perturbation contact network computed between all backbone NH groups and the rest of the protein, reflecting changes in the local environment around the amide protons. This outcome again supports the equivalence of the simulated ensembles with respect to those examined in NMR experiments. Thus, we inspected closely the dynamical behavior of the residues with the greatest change in temperature coefficient such as $f$K60, $f$L63, and $f$D14. (Fig. 4c). Residues $f$L63 and $f$D14 are also residues displaying curvature in their ¹H chemical shift profiles over temperature, indicative of their conformational flexibility.

Residues $f$K60 and $f$L63 are located in $f$α2 and are part of an altered salt bridge interaction (allosteric) network with two other helices, i.e., $f$α3 and $h$α1, which undergoes rearrangement upon effector binding[46]. Here, $f$L63, a hydrophobic residue, cannot be directly involved in the salt bridge network alteration, but is modulated through its neighbors. Upon temperature increase, the salt bridge between residues $f$K60 and $f$E90 breaks while that between $f$K60 and $f$E64 forms. Overall, this change produces a partial refolding in the lower end of the $f$α2 helix and the backbone H-bond between residue $f$L63-NH and $f$R59-CO becomes less stable upon temperature increase (see Fig. 4d–f and Supplementary Fig. 8). The presence of this backbone H-bond is consistent with the positive temperature coefficient of the $f$L63 residue recorded experimentally. The weakening of this bond with temperature increase explains the reduction of its slope around 50 °C with respect to that around 30 °C. In contrast, residue $f$D14 is located after the end of $f$β1 and its amide proton is not involved in secondary structure formation and largely exposed to solvent, in line with its negative temperature coefficient. However, we found that the $f$D14-NH can make an H-bond with the solvent or with the sidechain of residue $f$T53, located at the end of the $f$β2 sheet, both in *apo30* and *apo50*.

The distribution of $f$D14-$f$T53 H-bond distances over time (Supplementary Fig. 15) suggests such interaction is occurring more often in *apo50* than in *apo30*, indicating more frequent exchange of H-bond acceptor (water molecule or $f$T53 sidechain) with a temperature increase. Such dynamics are in line with the experimental observation of the temperature coefficient decrease at around 50 °C with respect to that around 30 °C.

In summary, the majority of notable changes in temperature coefficients between 30 °C and 50 °C are located near the effector site or at *sideR*, along the allosteric pathways in HisF. A complementary observation supporting this finding is provided by the change in HBs at the interface upon temperature increase and effector binding (Supplementary Fig. 16). The HB patterns once again evidence the strong structural parallelism between temperature increase from 30 °C to 50 °C and PRFAR binding at 30 °C.

Altogether our findings reveal that "local" temperature coefficients can be suitable for investigating allosteric key spots in temperature-activated systems, here corroborating the similarities between temperature increase and effector binding dynamics in IGPS.

## Temperature induced adaptation of essential dynamics
After showing that temperature increase produces contact changes that mimic those induced by PRFAR, the next question that we aimed to address is what are the specific protein motions that allow the formation of these allosteric contacts under different conditions. A simple and widely used technique to this scope is the analysis of principal components (PCA), which enables finding the most relevant collective variables defining the dynamics of a system. The method consists in obtaining the eigenvectors of the covariance matrix of atomic displacements, and then, although it is not formally guaranteed, the standard assumption is that the functional dynamics of the system is encoded in the leading eigenvectors[61]. Ordering the eigenvalues of the transformation decreasingly, it has been shown that a large part of the system's fluctuations can be described by the first few principal components, where the components associated to the largest eigenvalues usually incorporate a significant fraction of the variance. The ten eigenvectors corresponding to the largest eigenvalues computed for *apo30, holo30*, and *apo50* dynamics were comparable in each of the different replicas (see Supplementary Fig. 17) allowing us to focus on a representative set of simulations for each state. We project each 1 μs trajectory onto the respective largest principal component, to visualize the amplitude of fluctuation around the mean described by the largest collective mode. The overlay of the backbone coordinates (100 trajectory steps collected every 10 ns of each trajectory) is shown as spheres, colored in yellow to blue according to the time axis in Fig. 5a–c. In agreement with previous studies, we find that the first PC of the room-temperature apo trajectory describes the characteristic interdomain hinge-like (breathing) motion[45,46,49,62] that was also observed experimentally and recognized as important for allosteric signaling[63]. This motion changes in presence of PRFAR at 30 °C into a twist of one subunit with respect to the other.

As shown in Fig. 2b, the large displacement of loop1, present in *apo30*, is absent in *holo30*, supporting the idea that PRFAR transmits a specific signal whose effect is instantaneously received over the entire protein, resulting in a change in the profile of collective motion substantially different from the apo state. The increase of the temperature, instead, results in a milder effect, which combines elements of both the *apo30* and *holo30* dynamics. The hinge motion is not fully blocked, and the interdomain twisting motion is combined with an "uncontrolled" displacement of loop1. The projection of the three trajectories on the mean first and second PCs reveals additional insights at the level of loop1 (Supplementary Fig. 18). In *holo30* the combined components result in a very localized twist around residue $f$R59, while the temperature increase results in more delocalized fluctuation that involve residue $f$R59 as well as the $f$α6-$f$α6' loop and loop1. In line with these observations, the *apo50* essential dynamics carries the largest kinetic variance (Fig. 5d), with an increased gap in the magnitude of the second and third largest eigenvalues (Fig. 5f). Overall, all *apo50* recovers nearly as much of the cumulative variance in the first three components as *holo30*, which amounts to nearly 50% of the total, above that contained by the first three components in *apo30* (Fig. 5e).

To better analyze the similarity/difference between the first three components across the various simulations, we recomputed the transformation on the concatenated set of coordinates from each of the three trajectories, using a consistent eigen-basis for the principal components. In PC space (Fig. 5f) *apo30* and *holo30* appear as very distinct and mix only at the level of the second and third components. *Apo50*, instead, is very similar to *apo30* in PC1/PC3 space (Fig. 5h) and shows substantial overlap with both *apo30* and *holo30* in PC1/PC2 and PC2/PC3 configurations (Fig. 5g, i), suggesting again that the temperature partly recovers the dynamics exerted by the effector, compensating its absence by an adaptation of its global dynamics. In Fig. 5g–i, the regions of *apo50* that deviate both from *apo30* or *holo30* are those that carry the "adapted" behavior only emerging in *apo50*.

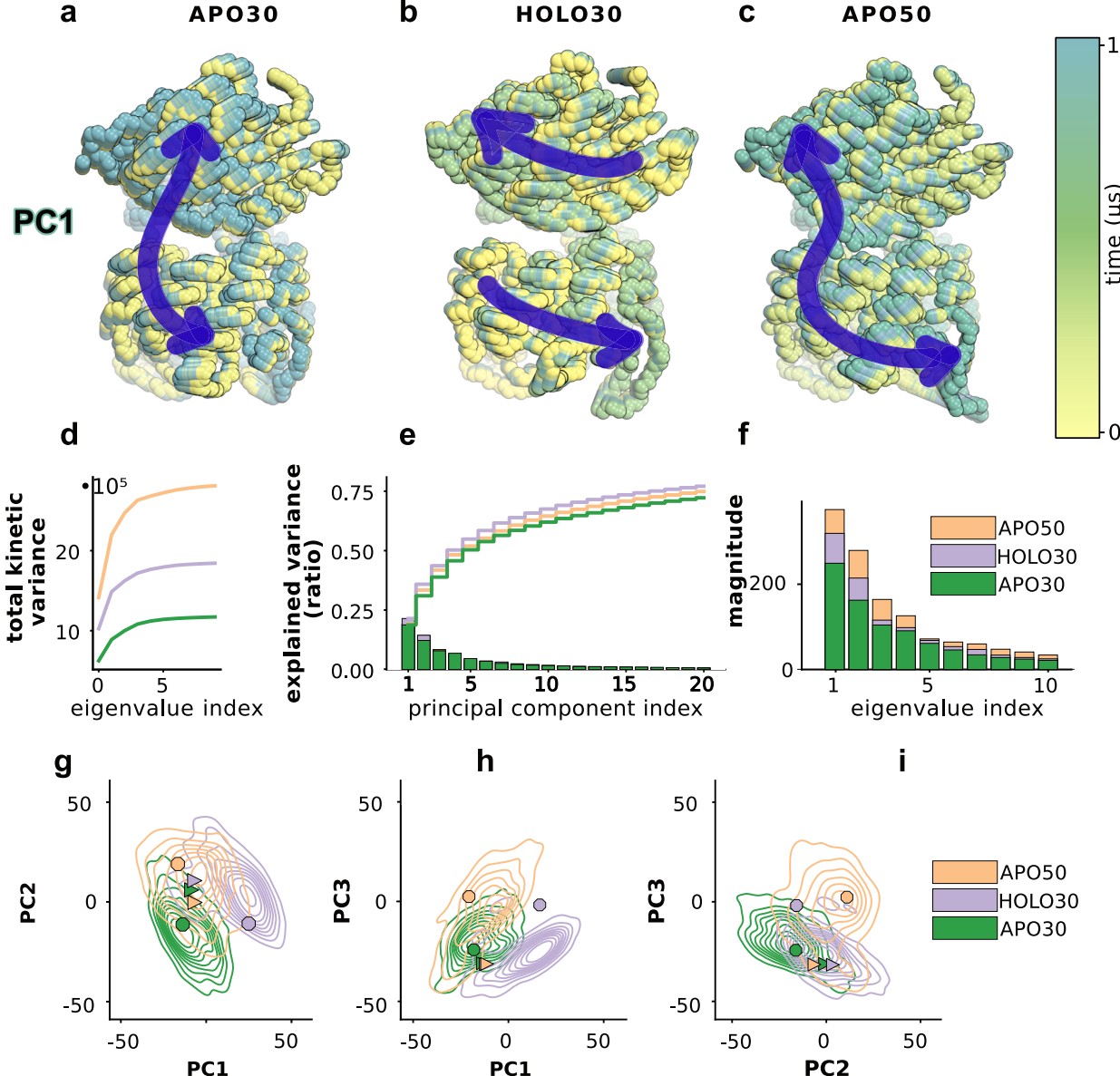

**Fig. 5 | Collective protein motions.** Essential motions of the *apo30* (**a**), *holo30* (**b**), and *apo50* (**c**) trajectories obtained by projecting each trajectory onto the first the principal component. The overlay of 100 structures (one every 10 ns) reproduces the motion described by the principal component over 1 μs time of each trajectory. The essential dynamics differ: *apo30* and *holo30* describe two very different motions, the first being the characteristic hinge motion, which changes into a twist of one domain relative to the other. The collective motions described by PC1 in *apo50* are a combination of that of *apo30* and *holo30*. **d** Total kinetic variance described by the first 10 components, computed as the cumulative sum of the square entries of the covariance eigenvalues. **e** Variance explained by the largest twenty (sorted) principal components. **f** Magnitude of the ten largest (sorted) eigenvalues. **g**–**i** Space described by the first three principal components of the *apo30*, *holo30* and *apo50* trajectories. Source data are provided as a Source Data file.

We have demonstrated that a temperature increase from 30 °C to 50 °C in the apo state of IGPS can activate a structural and dynamical pattern that remarkably resembles the effector-induced allosteric activation. By identifying the residues that belong to the signaling pathway, we have shown that both binding of the PRFAR effector or increasing temperature results in an activation of an external communication channel composed of solvent-exposed residues. In agreement with this, the perturbation of the residue contacts due to both temperature increase and PRFAR binding involves mainly solvent-exposed residues at the protein surface. Furthermore, in both cases the majority of contact alterations belong to the *sideR* of the protein, as demonstrated by our NMR temperature coefficient results and our eigenvector centrality analysis of microsecond dynamics. The presence of these similarities suggests that point mutations, such as V12A,

K19A, V48A, and D98A successfully implemented for altering dramatically the PRFAR-induced allostery[57], could be tested to evaluate their impact on the temperature-induced effect. Moreover, some residues (e.g., G15, T21, A54, V107, N109, A117, Q118, K162, L215, L222, K242) could be mutated to evaluate their implications in the temperature-induced allostery.

The main structural and dynamical differences between the thermal and PRFAR activation, are located in the proximities of the effector binding pocket, where the thermal fluctuations cannot mimic the specific directional interactions caused by the presence of PRFAR.

Collective motions are adapted and instrumental to ensure the formation of the relevant contacts that allow the allosteric signal to be transmitted across the two subunits: in presence of PRFAR the interface is closer than in the apo state and the characteristic opening/

closure hinge motion[45], is replaced by a twist of one subunit with respect to the other. At 50 °C in the absence of the ligand-binding signal—the activation follows an alternative route which involves a coordinated spring motion that combines both collective motions of apo and holo at 30 °C. This observation is not dissimilar to what we observed in a previous work on the mesophile yeast IGPS, where we found that the yeast analog adopts an alternative allosteric signaling pathway devoid of the entropic control of the interface opening, for optimum functionality at room temperature[45]. Elevated temperature and effector binding trigger similar contact changes, mostly located near the effector site or at *sideR*, along the allosteric pathways in HisF.

Overall, the results presented here provide fundamental insights on the allosteric activation at elevated temperatures. In comparison, allosteric activation induced by PRFAR shows less effect on protein collective motions, yet reminiscent of contacts among critical residues that enable allosteric signaling transfer. This observation is consistent with previous kinetic data that showed a sizable temperature-dependent increase of the intrinsic enzymatic activity of IGPS, although milder than that induced by the effector[29]. In this context, it could be speculated that the entropically-driven PRFAR binding[29] represents an evolutionary adaptation strategy to high temperatures by compensating the loss of PRFAR-induced activation with an increased PRFAR binding affinity. However, it is not clear what the physiological needs of *T. maritima* are at its optimal growth temperature versus ambient temperature and thus our speculation should stimulate new studies in this direction. Overall, this study opens the doors for the development of novel tools to control IGPS activity, such as rationally designed allosteric drugs, antipathogens, as well as new engineered variants.

## Methods

We apply a combination of computational methods based on molecular dynamics (MD) simulations, network theory correlation analysis techniques, and nuclear magnetic resonance (NMR)[45,46,49,62] to study temperature-dependent allosteric communication in *T. maritima* IGPS. Similar computational methods have been previously applied to investigate communication pathways and allostery in proteins and protein-tRNA/DNA molecular systems[64–76].

### Molecular dynamics simulations

The structural models for apo and holo forms of IGPS were based on the crystal structure of *T. Maritima* IGPS (PDB ID 1GPW, 2.4 Å resolution)[76,77]. In this structure, chain C has the particularity to possess loop1 in a conformation prone to effector binding, thus we built the apo structure by extracting chains C and D of the HisH-HisF complex and reversing the D11N engineered mutation back to its wild-type form. The PRFAR-bound structure was built as described previously[46]. The protein–ligand complex was parameterized with the CHARMM36m[78,79] and the generalized CHARMM force fields[80] using the CHARMM-GUI[81]. We kept all water molecules associated with the two chains and solvated the structures by using the explicit TIP3P model[76] to obtain a cubic box. The protein was placed at the center of a 110 × 110 × 110 Å box with a distance of at least 10 Å from the box edges (~20,000 water molecules). Cl⁻ and Na+ ions were added at randomized positions in the box up to neutralization of the total charge.

We used AmberTools2021[82] to convert the CHARMM file format to Amber, and the AmberGPU[83,84] package with the CHARMM36m force field for subsequent minimizations heating, and production runs, (further details of pre-equilibration procedure and MD production runs are described in Supporting Information, SI). To compare the effect of PRFAR and of the increase in temperature, we simulated the apo and the IGPS-PRFAR (holo) structures at 30 °C and 50 °C. We will refer to the simulations of the apo system performed at 30 °C and 50 °C as *apo30* and *apo50*, respectively, and *holo30* and *holo50* for the holo IGPS protein complexes. We simulated each system for 1.5 µs and extracted ~10,000 equally spaced frames from the last 1 µs of each trajectory for analysis reported in the main text. Furthermore, we verified the statistical relevance of the simulated ensembles by running three additional replicas (replicas 1,2,3) of each state. Each of the three replicas trajectories of all states were simulated for production runs (i.e., after pre-equilibration) of 1.2 µs. Configurations from the last µs of each replica were sampled extracting one frame every 100 ps, resulting in 10,000 frames for each system, which were used for comparison to the original simulations (see SI).

We postprocessed and analyzed the trajectories using MDTraj[85], CPPTRAJ[86] and pytraj[86]. The secondary structure analysis was performed by using a dictionary for the secondary structure of proteins (DSSP)[87] as implemented in MDTraj[85]. Throughout the trajectories, a residue was assigned to one of the following secondary structure elements: helix (either α-helix, 3-helix, or 5-helix), sheet (extended strand, isolated β-bridge), or coil (turn, bend or loop and irregular elements). The SSE persistency, defined as the percentage of frames in which each residue retains the same secondary structure state as in the crystal structure, was assigned relative to the secondary structure assignments performed on the 1GPW crystal structure.

### Eigenvector centrality analysis

In order to elucidate the allosteric pathways and pinpoint the changes in IGPS dynamics upon PRFAR binding and temperature increase, we employed the eigenvector centrality (EC) analysis recently developed by our group[44]. The method relies on mapping the MD trajectory into a graph composed of nodes separated by edges. Each node in the graph represents an α-carbon of a given amino acid.

The EC method is a powerful tool that allows us to identify the key residues involved in the allosteric pathways and to understand the changes in IGPS dynamics upon PRFAR binding and temperature increase. Edges between nodes are defined through an adjacency matrix **A**, where $\mathbf{A}_{ij}$ is the generalized correlation coefficient $r_{MI}$ between nodes $i$ and $j$, given by:

$$r_{MI}\left[x_i, x_j\right] = \left(1 - \exp\left(\frac{-2}{3}I\left[x_i, x_j\right]\right)\right)^{1/2}, \quad (1)$$

where $I[x_i, x_j]$ represents the mutual information between the two amino acids[75], and is computed as

$$I\left[x_i, x_j\right] = S\left[x_i\right] + S\left[x_j\right] - S\left[x_i, x_j\right] \quad (2)$$

$$S\left[x_i\right] = -\int dx_i\, p\left[x_i\right]\ln\left(p\left[x_i\right]\right) \quad (3)$$

$$S\left[x_i, x_j\right] = -\iint dx_i\, dx_j p\left(\left[x_i, x_j\right]\right)\ln\left(p\left[x_i, x_j\right]\right), \quad (4)$$

where $S[x_i]$ and $S[x_i, x_j]$ are the marginal and joint Shannon entropies, respectively[75] while $p[x_i]$ and $p([x_i, x_j])$ are probabilities of atomic displacement computed over thermal fluctuations sampled by MD simulations at equilibrium. The generalized correlation coefficient $r_{MI}$ ranges from 0 (for uncorrelated variables) to 1 (for fully correlated variables).

Once the adjacency matrix is obtained, diagonalizing the matrix provides an eigenvector whose values are related to each residue. The EC of an amino acid, $c_i$, can be defined as the weighted sum of the EC's of all the residues connected to it by an edge, $A_{ij}$:

$$c_i = \varepsilon^{-1}\sum_{j=1}^{n} \mathbf{A}_{ij}c_j, \quad (5)$$

where **c** is the eigenvector of **A** corresponding to the leading eigenvalue $\varepsilon$. The EC metric provides a measure of how well-connected each node is to other well-connected nodes in the network. This notion of eigenvector centrality allows for recognition of patterns of dynamical changes associated with PRFAR binding in IGPS[44] and is here used to define and compare those associated with temperature increase. Notably, if normalized, the EC coefficients of different states can be used to reliably compare different networks[44].

## Optimal pathways for motion transmission

We studied the optimal pathways for motion transfer between the active site (*h*C84, *f*H178, *f*E180) and the effector binding site to understand how the cross-talk between them is altered by a temperature increase and by the PRFAR binding process. To this aim, we employed the Dijkstra algorithm[88], designed to find the roads that minimize the total distance traveled. In this study, the inter-node distance was defined as:

$$w_{ij}^{(0)} = -\log\left[r_{MI}\left[x_i, x_j\right]\right]. \tag{6}$$

Therefore, the minimization of the total **w** traveled is equivalent to a maximal correlation between the initial and the final nodes of the path. The algorithm begins by defining starting and destination nodes. In our case, the starting nodes are the residues featuring hydrogen bonds (HB) with the PRFAR phosphate groups in the holo form, and the destination node is *h*C84/*h*G50 in the active site (where the glutamine substrate binds). The pathway from the former to the latter is optimized iteratively, yielding the pathways that minimize the total distance (and therefore maximize the correlation).

The cross-talk between two amino acids does not necessarily occur exclusively through the optimal path. In this study, we built pathways merging the 50 sub-optimal paths, representing the most likely pathways of motion transmission between the active and the effector sites. This is important to note because many sub-optimal paths with similar influence might contribute to the communication between distant residues.

## Perturbation contact networks analysis

In order to determine the degree to which the effector binding causes effects that differ from those elicited by temperature increase, we applied the dynamical perturbation contact network (DPCN) analysis method recently proposed by our group[89]. We have previously performed the DPCN to monitor the PRFAR binding effects on amino acid residue contacts[89] and here we compare it with the temperature effect on the *apo30* system by determining the contact changes with respect to the *apo50* enzyme.

Each protein-weighted contact network is built by assigning a weight $w_{ij}$ to each edge (linking the *i*-th and *j*-th residues) that is the number of contacts between the residues. The contact condition is here defined for each pair of residues when it exists in two atoms (at least one per residue) whose distance is below a given distance cutoff (here set to 5 Å) for each snapshot extracted from the MD trajectories (i.e., 10,000 snapshots for each system). Further computational details can be found in the reference work on the effector binding DPCN[89]. To allow easy visual inspection of DPCN results in the manuscript, the edges are colored in red if PRFAR binding or a temperature increase induces an increase in weight ($\Delta w_{ij} > 0$), and in blue if instead the contact number is reduced ($\Delta w_{ij} < 0$). Moreover, just for visualization purpose, a weight threshold ($w_t = 6$) is applied so that only the edges with $|w_{ij}| > w_t$ are depicted. Here, atomic contacts are computed including all heavy atoms (i.e., excluding hydrogens) of the amino acid residues (symmetric contacts) and for each residue, including atoms from its backbone amide (NH) group in contact with the heavy atoms of rest of the protein (asymmetric contacts). The obtained DPCNs are provided as SI files.

## Principal components analysis

PCA was performed employing the *sklearn.decomposition*.PCA function in the Scikit-learn library using python8.3.6[90]. First, all simulations were aligned with MDAnalysis[91] to the crystal structure coordinates using C$\alpha$ atoms. Next, we removed instantaneous linear correlations among the coordinates within each trajectory by applying a linear transformation that diagonalizes the respective covariance matrix of backbone atomic displacements. We project the original backbone trajectories onto the first principal components (sorted eigenvectors) and visualize their motion. In order to compare the essential dynamics of three systems (*apo30, holo30,* and *apo50*) in their low-dimensional free energy landscapes, we repeated the procedure using the concatenated coordinates of each trajectory to fit a single transformation function. The fitted transformation function was then applied to reduce the dimensionality of each system's simulated backbone coordinates.

## Predicted NMR chemical shifts

The structures sampled in the MD simulations were used to predict the backbone $^1H^N$, $^{15}N^H$, $^{13}C\alpha$, and $^{13}C'$ NMR chemical shifts using the SHIFTX2 method[55]. This program combines ensemble machine learning techniques with sequence alignment-based methods and its algorithm has been tested with high-resolution X-ray structures with verified chemical shift assignments. The SHIFTX2[55] analysis was performed on 10,000 configurations from each 1 μs MD simulation to extract the backbone $^1H^N$, $^{15}N^H$, $^{13}C\alpha$, and $^{13}C'$ chemical shifts at 30 °C and at 50 °C; these results were compared to experimental results.

## Experimental NMR chemical shift

To obtain $^{13}C\alpha$ and C' chemical shift values for residues in the HisF subunit of IGPS, the TROSY-based HNCA and HNCO triple resonance experiments[92,93] were acquired for the wild-type enzyme. The HisF subunit was isotopically enriched with $^2H$,$^{13}C$, and $^{15}N$ isotopes, whereas HisH was grown in perdeuterated M9 minimal media with naturally abundant carbon and nitrogen, as described previously[57]. The 3D HNCA and HNCO experiments were acquired at 30 °C and 50 °C for the apoenzyme and for the effector analog IGP-bound holoenzyme at 30 °C. For the HNCA experiment, 16 scans were collected with 42 and 36 increments in the $t_1$ and $t_2$ indirect dimensions, respectively. The spectral width was set to 12,000, 4545, and 2000 Hz for $^1H$, $^{13}C$ and $^{15}N$; the recycle delay was set to 0.7 s. For the amide proton chemical shift temperature coefficient analysis, 2D $^1H$–$^{15}N$ TROSY spectra were acquired with 32 scans and 128 increments in the $t_1$ dimension with corresponding spectral widths of 12,000 Hz and 2800 Hz and a 1 s recycle delay. The temperature was calibrated using an external methanol sample. Amide proton and nitrogen chemical shifts were obtained from the 2D HSQC experiments recorded at 6 temperatures ranging from 20 to 50 °C (20, 25, 30, 35, 40, and 50 °C). Chemical shifts were referenced using DSS as an internal standard with the $^1H$ resonance frequency of DSS set to 0 ppm. All 2D and 3D NMR data were collected at a static magnetic field strength of 14.1 T (600 MHz) on a Varian Inova instrument, processed in NMRPipe[94] and analyzed with Sparky[95]. Temperature coefficients for individual residues were further analyzed and visualized using GraphPad Prism 9.0.

## Reporting summary

Further information on research design is available in the Nature Portfolio Reporting Summary linked to this article.

# Data availability

The structures reported in this work were generated from the crystal structure accessible under the PDB accession code 1GPW. The PDB structures corresponding to the first frame of each trajectory are also provided in the PDB folder within the GitHub repository, https://github.com/Batista-Chemistry-Lab/heatIGPS. The NMR data used in

this study are deposited in the BMRB database under accession code BRMB 15741 ($^1$H, $^{15}$N and $^{13}$C resonance assignment of imidazole glycerol phosphate (IGP) synthase protein HisF from *Thermotoga maritima*). The source data of the figures are provided in the Source Data file. Source data are provided with this paper.

## Code availability
Codes to reproduce the PCA, DPCN, correlation, pathways analysis and secondary structures analysis are made available at the Github link: https://github.com/Batista-Chemistry-Lab/heatIGPS.

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

## Acknowledgements
J.P.L. and V.S.B. acknowledge support from NIH GM R01-106121. This work was also supported by a generous allocation of high-performance computing time from NERSC. I.R. gratefully acknowledges the use of HPC resources of the "Pôle Scientifique de Modélisation Numérique" (PSMN) of the ENS-Lyon, France.

## Author contributions
U.N.M, J.P.L., I.R., V.S.B. designed the project, F.M. and F.T. performed molecular dynamics simulations, F.M., U.N.M., A.G., F.T., analyzed the data, A.C. performed and analyzed NMR experiments. G.W.K., P.N., contributed to secondary structure and hydrogen bonds analysis shown in supporting information. B.B. provided part of computational resources. F.M., U.N.M., A.G., A.C. wrote the initial draft. All authors contributed to discussions and revisions of the manuscript.

## Competing interests
The authors declare no competing interest.
