## [Peer Review File · Nature Communications]

REVIEWER COMMENTS

Reviewer #1 (Remarks to the Author):

This manuscript focuses on how temperature modulates the collective dynamics underlying allostery in imidazole glycerol phosphate synthase (IGPS) and on how such modulations resemble those caused by ligand (PRFAR) binding. The primary tools are 1.5 μ s MD simulations (residue centrality, information pathways, DPCN analysis) that were validated through comparisons between calculated and measured Ca ppm values and by the non-linear temperature-dependence of ^1H ppm values.

The resulting atomistic picture reveals that temperature may act as an important, yet previously overlooked, allosteric modulator that resembles the PRFAR ligand with respect to changes in local dynamics as opposed to collective motions, which are instead adaptive. The authors also identify the allosteric pathways controlled by temperature and PRFAR, which are composed primarily of solvent-exposed residues. These results could have transformative implications for our understanding of allostery which is traditionally thought to be triggered primarily by ligands alone.

Comments:

- 1) The authors mention unbinding events occurring for the PRFAR:IGPS complex at 50 oC but no evidence or further discussion is provided. Such unbinding should be quantified for example reporting the distance between the two centers of mass as a function of simulated time in the MD trajectory.
- 2) The ability to differentiate sideL and sideR is critical to relate the MD results to the NMR data. However, the 7AC8 structure of IGPS does look pseudo-symmetric, suggesting potential overlap between at least some the sideL and sideR NMR peaks. Could this be explained further through supplementary text and figures illustrating differences and similarities between the two sides?
- 3) Could the authors include positive and/or negative controls of the primary results? For example, based on the signaling pathways they elucidated, is it possible to identify mutations that amplify or reduce the temperature-induced activation of IGPS?
- 4) Can the authors articulate further the thermodynamical and physiological implications of the temperature-induced activation of apo IGPS? Does this mean that IGPS is entropically driven? Does the T-induced constitutive activation of IGPS serve as an adaption mechanism?
- 5) Minor: What does green ribbon mean in Fig 1?

Reviewer #2 (Remarks to the Author):

This manuscript describes the combined experimental and computational investigation of temperature or ligand-induced allosteric signaling in imidazole glycerol phosphate synthase. The combined results indicate that increase in temperature induces similar allosteric activation as effector binding. The observed allosteric activation is due to changes in the structure of specific surface residues, and concomitant dynamical effects that affect the essential dynamics of the protein as observed by experimental temperature coefficient changes and network correlation analyses. Overall, the results present an interesting example and explanation of allosteric control and alternative approaches to induce it. However, there are several issues that need to be addressed before this manuscript can be published:

-There are a large number of missing details about the computational methods including (but not limited to) what is the size of the box and distance between protein surface and box edge; if or how the charge of the systems was neutralized and/or if the ionic strength of the systems was considered. Specifics of the MD runs (i.e. number of minimization steps, ensemble, timestep, equilibration method, long-range interaction method, etc) are missing.

-Only a single MD trajectory per system is reported, it would be better to perform at least two more simulations per system to determine if the results are consistent.

-Figure 4 E-G, It would be more informative if the average distance and std. dev. for the H bond between L63 and R59 would be included. How is this non-bonded interaction defined? is the H bond between an atom on the side-chain of R and an atom in the backbone of L?

-Figure 5: caption is incomplete

Reviewer #3 (Remarks to the Author):

The manuscript by Maschietto et al. investigates the correlation among the effect of temperature and the presence of an allosteric effector on the dynamics of the Imidazole Glycerol Phosphate Synthase from a thermophilic organism (*T. maritima*).

The work is based on the methodology previously developed and already applied to the same enzyme to unveil the allosteric communication path. The novelty of the present work is the investigation of the effect of temperature. In fact, for this thermophilic enzyme, the allosteric activation is weak at the functional working temperature of the protein as compared to ambient conditions (see Ref. 29 in the manuscript). The authors show that the thermal perturbation on the protein dynamics and flexibility, and network of interactions, is similar to what induced by the effector PRFAR at ambient condition. This explains why the presence of PRFAR at the working condition of the enzyme has a much weaker effect on the enzyme catalytic efficiency.

I find this work interesting since it adds a new piece of information on the relationship among allostery and thermal excitation. The conclusions are adequately supported but extra information (see below) is needed to improve the manuscript and clarify better some aspects.

I have some points that must be considered before publication.

1. The work focuses only on the allosteric pathway, and the long range communication. However, the boost of the catalytic power should also be considered in term of reorganisation of the active site. The authors should characterise how the presence of the allosteric effector and the higher functional temperature modulate the organisation of the catalytic site. How the global reorganisation induced by the effector, or the temperature, translates in a higher catalytic efficiency? How the chemical step of the reaction is affected? Ideally, a reader would expect information on the different energetics of the catalytic step for the considered perturbations (PRFAR and temperature).
2. The authors should compare whether the global structure of the APO enzyme at high temperature gets closer to the structure of the Holo (with PRFAR) enzyme at ambient condition by computing the time evolution of the RMSD of the dynamics of the former with respect to the equilibrated structure of the latter. One should expect a decreases over time. The analysis can be extended to the active site, and also considering structures with the presence of the substrate in the catalytic site (see point 1).

3. In the methodology “Eigen Vector Centrality Analysis” it could be useful to stress that the analysis performed on different structures can be directly compared (see results in Fig. 2 Panel a).

4. Binding of PRFAR, pg 7. The authors mention that the Holo structure of the protein is unstable at 50 C and PRFAR unbinds. This must be taken with a grain of salt in view of possible force field artefacts at high temperature. The authors should try to constrain the effector in the binding pocket. Moreover this seems in contradiction with what reported in the Conclusion about the experimental evidence of increased binding affinity of PRFAR at high temperature (if I understand correctly).

Also the explanation of the evolutionary gain due to PRFAR increased binding affinity that compensates its weak allosteric activation role at high T is not really clear to me (see Conclusion). If important it must be explained and developed.

5. The investigation of the temperature effect on allostery has been investigated also by others in the context of the malate/lactate dehydrogenase protein families. It was showed how temperature mimics the effect of an allosteric effector, and can play a functional regulation role. See: J. Phys. Chem. B (2020) 124, 6, 1001–1008, and Sci. Reports. (2016) 7, 41092.

We thank you and the reviewers for the comments and suggestions. We indicate the comments from the reviewers in green and our answers in black. Parts extracted from the main text are indicated in *italics*. Furthermore, we have highlighted the revised version of the manuscript to indicate in blue changes with respect to the original manuscript.

Answers to Reviewers

Reviewer #1 (Remarks to the Author):

This manuscript focuses on how temperature modulates the collective dynamics underlying allostery in imidazole glycerol phosphate synthase (IGPS) and on how such modulations resemble those caused by ligand (PRFAR) binding. The primary tools are 1.5 μ s MD simulations (residue centrality, information pathways, DPCN analysis) that were validated through comparisons between calculated and measured Ca ppm values and by the non-linear temperature-dependence of 1H ppm values.

The resulting atomistic picture reveals that temperature may act as an important, yet previously overlooked, allosteric modulator that resembles the PRFAR ligand with respect to changes in local dynamics as opposed to collective motions, which are instead adaptive. The authors also identify the allosteric pathways controlled by temperature and PRFAR, which are composed primarily of solvent-exposed residues. These results could have transformative implications for our understanding of allostery which is traditionally thought to be triggered primarily by ligands alone.

We really thank the Reviewer for appreciating the novelty and transformative implications of our work.

Comments:

1) The authors mention unbinding events occurring for the PRFAR:IGPS complex at 50°C but no evidence or further discussion is provided. Such unbinding should be quantified for example reporting the distance between the two centers of mass as a function of simulated time in the MD trajectory.

We thank the Reviewer for this remark. We have performed the suggested analysis, which allowed us to restate more accurately our conclusions on the unbinding 'events'. As indicated by the Reviewer, our analysis is based on the unbinding 'dynamics'.

In order to improve the overall statistical analysis in our work (following the suggestions of Reviewer #2), we performed three new 1 μ s simulation replicas of (also) *holo50*, providing enough statistics to analyze the unbinding dynamics. We computed the relative distance between the center of mass of PRFAR and the effector site (i.e., protein residues located in a sphere radius of 6 Å from PRFAR) along the trajectories, as suggested by the Reviewer and we reported the results in a new Figure in the SI (Figure S11, also shown below). At 50°C, the distance between the two centers of mass varies greatly across the trajectories, suggesting that PRFAR explores different positions, in contrast to what occurs at 30°C (see Figure S11A). Moreover, the minimum distance between any atom of the protein and PRFAR remains within hydrogen bond (HB) distance for most of the simulations. The PRFAR dynamics in *holo50* features molecular interactions with different residues, holding the effector bound to the protein while not necessary in its binding pocket. Only very rarely, the effector features broken HBs (*min d* > 2 Å) with the HisF protein (see Figure S11A bottom). As reported in the new Figure S11C, we found that the number of hydrogen bonds per frame in the *holo50* simulations is comparable to that of *apo30*. The number of residues involved in hydrogen bonds, however, increases at elevated temperature conditions. The results thus indicate increased mobility of PRFAR at higher temperature, leading to significant displacements in the pocket.

Figure SI1

To account for these findings, we have modified the main text as follows:

Page 6:

*This behavior is also in agreement with the observed **increased mobility** of PRFAR in the holo50 simulations, leading to **significant displacements in the pocket** that were **almost absent** in the same timescales in the room temperature dynamics (SI, Figure SI1 and associated discussion).*

In the SI, we have also added the discussion of the new Fig. SI1, as follows:

SI, Page 6:

To improve the overall statistics, we performed four 1 μ s simulation replicas of (also) holo50, providing enough statistics to analyze the unbinding dynamics. We computed the relative distance between the center of mass of PRFAR and the effector site (i.e. protein residues located in a sphere radius of 6 Å from PRFAR) along the trajectories, and we reported the results in Figure SI1. At 50°C, the distance between the two centers of mass varies greatly across the trajectories, suggesting that PRFAR explores different positions, in contrast to what occurs at 30°C (see Figure SI1A). PRFAR dynamics in holo50 features molecular interactions with different residues, holding the effector bound to the protein while not necessarily in its binding pocket. Only very rarely, the effector features broken HBs ($min d > 2$ Å) with the HisF protein (see Figure SI1A bottom). As reported in the new Figure SI1C, we found that the number of hydrogen bonds per frame in the holo50 simulations is comparable to that of apo30. The number of residues involved in hydrogen bonds, however, increases at elevated temperature conditions. The results

thus indicated increased mobility of PRFAR at higher temperature, leading to significant displacements in the pocket.

2) The ability to differentiate sideL and sideR is critical to relate the MD results to the NMR data. However, the 7AC8 structure of IGPS does look pseudo-symmetric, suggesting potential overlap between at least some of the sideL and sideR NMR peaks. Could this be explained further through supplementary text and figures illustrating differences and similarities between the two sides?

We agree with the Reviewer that the current definition of sideL and sideR is not unequivocal. To avoid any confusion, we included a table in the SI (see Table SI1) listing separately HisF residues in sideL from those in side R. While the two sides are not symmetric this distinction is essential to elucidate the secondary structures that are mostly affected by either of the effectors (PRFAR, or temperature). It is worth noting that the apparent pseudo-symmetry of the structure has no impact on the ability to differentiate NMR resonances. Indeed, we observe distinct NMR peaks for each resonance and there is no confusion between resonances for peaks in SideL and SideR. The resonances were assigned by established triple resonance experiments.

3) Could the authors include positive and/or negative controls of the primary results? For example, based on the signaling pathways they elucidated, is it possible to identify mutations that amplify or reduce the temperature-induced activation of IGPS?

We really thank the Reviewer for this important comment.

Indeed, while it has been demonstrated that our simulations can provide effective suggestions for mutagenesis experiments, we did not mention that observation in the original version of the manuscript, and we did not include suggestions for point mutations that could affect the temperature-induced allosteric signaling. In order to comply with the request, we have modified the text as follows:

Page 15:

The presence of these similarities suggests that point mutations, such as V12A, K19A, V48A, and D98A successfully implemented for altering dramatically the PRFAR-induced allostery,⁸⁶ could be tested to evaluate their impact on the temperature-induced effect. Moreover, some residues (e.g., G15, T21, A54, V107, N109, A117, Q118, K162, L215, L222, K242) could be mutated to evaluate their implications in the temperature-induced allostery.

4) Can the authors articulate further the thermodynamical and physiological implications of the temperature-induced activation of apo IGPS? Does this mean that IGPS is entropically driven? Does the T-induced constitutive activation of IGPS serve as an adaptation mechanism?

We thank the Reviewer for making this relevant comment. At the moment, it is not clear what are the physiological needs of *T. Maritima* at its optimal growth temperature *versus* ambient temperature. So, we have revised the manuscript to indicate that our comments on that aspect remain speculative. Still, our experimental data does show that PRFAR binding is entropically-driven (ref. 86, Lipchock and Loria, Structure v.18, pp1596-1607, 2010) with $\Delta S \sim -50$ cal/mol/deg, whereas Gln binding is slightly entropically favored ($\Delta S = 8.92$ cal/mol/deg).

Since it was not mentioned in the main text, we specified the entropically-driven binding of PRFAR as follows:

Page 2:

The allosteric ligand *N*'-[(5'-phosphoribulosyl)formimino]-5-aminoimidazole-4-carboxamide-ribonucleotide (PRFAR) features an entropically-driven binding to the HisF subunit, enhancing glutamine hydrolysis at room temperature 5000-fold over its basal level.

Moreover, we modified the Conclusions as follows:

Page 15:

Overall, the results presented here provide fundamental insights on the allosteric activation at elevated temperatures. In comparison, allosteric activation induced by PRFAR shows less effect on protein collective motions, yet reminiscent of contacts among critical residues that enable allosteric signaling transfer. This observation is consistent with previous kinetic data that showed a sizable temperature-dependent increase of the intrinsic enzymatic activity of IGPS, although milder than that induced by the effector.²⁹ In this context, it could be speculated that the entropically-driven PRFAR binding²⁹ represents an evolutionary adaptation strategy to high temperatures by compensating the loss of PRFAR-induced activation with an increased PRFAR binding affinity. However, it is not clear what the physiological needs of *T. maritima* are at its optimal growth temperature versus ambient temperature and thus our speculation should stimulate new studies in this direction. Overall, this study opens the doors for the development of novel tools to control IGPS activity, such as rationally designed allosteric drugs, antipathogens, as well as new engineered variants.

5) Minor: What does green ribbon mean in Fig 1?

In Figure 1, the protein structure is colored using a spectral color scheme, where the different secondary structures appear colored with a gradient fashion. To make this clear, we have changed the caption of Figure 1, as follows:

Page 2:

Figure 1. Molecular representation of IGPS. IGPS is a bienzyme composed of two subunits, i.e. HisF and HisH, that constitute the cyclase and glutaminase domains, here colored respectively in green-to-blue and red-to-yellow gradients, respectively, and separated by a dotted line which marks the interface between HisF and HisH. The labels (α 2, α 3, β 2, loop1, α 1, Ω -loop) indicate secondary structure elements that are directly involved in the allosteric regulation, as found by previous studies.

Reviewer #2 (Remarks to the Author):

This manuscript describes the combined experimental and computational investigation of temperature or ligand-induced allosteric signaling in imidazole glycerol phosphate synthase. The combined results indicate that increase in temperature induces similar allosteric activation as effector binding. The observed allosteric activation is due to changes in the structure of specific surface residues, and concomitant dynamical effects that affect the essential dynamics of the protein as observed by experimental temperature coefficient changes and network correlation analyses. Overall, the results present an interesting example and explanation of allosteric control and alternative approaches to induce it.

We thank the Reviewer for the supporting comments.

However, there are several issues that need to be addressed before this manuscript can be published:

-There are a large number of missing details about the computational methods including (but not limited to) what is the size of the box and distance between protein surface and box edge; if or how the charge of the systems was neutralized and/or if the ionic strength of the systems was considered. Specifics of the MD runs (i.e. number of minimization steps, ensemble, timestep, equilibration method, long-range interaction method, etc) are missing.

We have revised the Supporting Information to provide a detailed the requested information. We have also modified the section “*Molecular dynamics simulations*” in “Materials and methods” of the main text as follows:

Page 3:

The structural models for apo and holo forms of IGPS were based on the crystal structure of T. Maritima IGPS (PDB ID 1GPW, 2.4 Å resolution). In this structure, chain C has the particularity to possess loop1 in a conformation prone to effector binding, thus we built the apo structure by extracting chains C and D of the HisH-HisF complex and reversing the D11N engineered mutation back to its wild-type form. The PRFAR-bound structure was built as described previously. The protein-ligand complex was parameterized with the CHARMM36m and the generalized CHARMM force fields using the CHARMM-GUI. We kept all water molecules associated with the two chains and solvated the structures by using the explicit TIP3P model to obtain a cubic box. The protein was placed at the center of a 110 x 110 x 110 Å box with a distance of at least 10 Å from the box edges (~20000 water molecules). Cl- and Na+ ions were added at randomized positions in the box up to neutralization of the total charge. We used AmberTools2021 to convert the CHARMM file format to Amber, and the AmberGPU package with the CHARMM36m force field for subsequent minimizations heating, and production runs, (further details of pre-equilibration procedure and MD production runs are described in Supporting Information).

-Only a single MD trajectory per system is reported, it would be better to perform at least two more simulations per system to determine if the results are consistent.

We agree with the Reviewer that checking the statistical relevance of the reported simulations is an important point. So, we have run three additional replicas for each state and presented RMSD and RMSF and generalized correlation coefficient distributions for replicas 1,2,3 and compared the latter to those obtained for the simulations discussed in the main text (replica 0). The results are shown in Figures S12-S14, revealing excellent agreement between the different replicas, with computed correlation values between C α -RMSF profile of replica 0 (discussed in the main text) and the three additional replicas (rep. 1, rep.2, rep. 3) all exceeding 0.9. A very good agreement is found also for the RMSD profiles, as confirmed by the substantial overlap in corresponding kernel density estimates. Furthermore, for each set of simulations (*apo30*, *holo30*, *apo50*) we checked that the distribution obtained from the squared sum of the first ten eigenvectors of the covariance matrix computed for each replica (0, 1, 2, 3) have significant overlap (SI, Figure S17). This in turn confirmed that the essential dynamics of the independent replicas is comparable.

We added several instances to the main text to refer to these comparisons, as follows:

In the “Materials and methods” section,

Page : 3

We simulated each system for 1.5 μ s (*replica 0*) and extracted ~10000 equally spaced frames from the last 1 μ s of each trajectory for analysis reported in the main text. *Furthermore, we verified the statistical relevance of the simulated ensembles by running three additional replicas (replicas 1,2,3) of each state. Each of the three replicas trajectories of all states were simulated for production runs (i.e., after pre-equilibration) of 1.2 μ s. Configurations from the last μ s of each replica were sampled extracting one*

frame every 100 ps, resulting in 10000 frames for each system, which were used for comparison to the original simulations (see SI).

In the Results section:

Page 8:

To ensure reproducibility and statistical relevance of our outcomes, we analyzed correlations and fluctuations for three additional replicas for each state (apo30, holo30 and apo50), shown in Figures SI3-5. The high correspondence of the additional replicas to the set of simulations presented in the main text is confirmed by small divergence in the RMSD profiles, with correlation of RMSF distributions for the three new replicas all exceeding 0.9.

We extended the analysis of secondary structure persistency changes (Figure 3 in main text) using the concatenated sets of apo30, holo30 and apo50 trajectories, which shows that a more extensive sampling reveals even greater similarities between the temperature-induced and effector-induced dynamics than those previously reported in the original manuscript. In order to keep showing the few differences between these two allosteric activations we decided to maintain the original Figure 3 in the main text, as it contains the comparison between the two replicas (replica 0) with the clearest differences. Still, we slightly revised the discussion of the results to account for the observations resulting from the average data of all four replicas (replicas 0-3) of each system as follows:

Page 10:

Aside from these similarities, temperature increase and PRFAR binding lead to only few different outcomes, some of which, as could be expected, in the region near the effector site. For the selected simulations compared in Figure 3 (replica 0), an increased helicity in fβ6-fa6 is observed in holo30 but almost absent in apo50. The formation of these helices is mostly triggered by the interactions with PRFAR (see Figure SI8). When considering additional replicas in the analysis, the average effect of temperature and PRFAR become even closer (SI, Figure SI9), supporting our hypothesis that the effect of temperature closely mimics that of PRFAR. The RMSF difference plots (SI, Figure SI7) also show an increase in stiffness in holo30 in the fβ6-fa6 and fβ7- fa7 turns near the effector site that is only barely present in apo50. The fβ8-fa8 region (lime green in Figure 3 and Figure SI9), instead, is slightly more affected by temperature than by the presence of PRFAR. Only a few residues are affected exclusively by a temperature increase when the average over all replicas was considered (SI, Figure SI9), including G15, T21, A54, V107, N109, A117, Q118, K162, L215, L222, K242.

The results including the new replicas trajectories were added also in the description of the temperature induced adaptation of essential dynamics, implementing the following changes in the text:

Page 13:

The 10 eigenvectors corresponding to the largest eigenvalues computed for apo30, holo30 and apo50 dynamics were comparable in each of the different replicas (see SI Figure SI17) allowing us to focus on a representative set of simulations (replica 0) for each state. We project each 1 μs trajectory ...

-Figure 4 E-G, It would be more informative if the average distance and std. dev. for the H bond between L63 and R59 would be included. How is this non-bonded interaction defined? Is the H bond between an atom on the side-chain of R and an atom in the backbone of L?

We have revised the manuscript to add the standard deviation and mean of each distribution in the Figure 4E-G and an explicit definition of the H-bond interaction, changing the caption accordingly:

Page 12:

D) Evolution of the fL63-fR59 backbone hydrogen bond (defined as the distance between the backbone atoms fL63-N and fR59-O) over time in the apo30, apo50 and holo30. E-G) Overlay of 100 configurations sampled (1 every ten ns) for the fL63 and fR59 residues along the apo30, holo30 and apo50 MD trajectories. The mean and standard deviation of the fL63-fR59 backbone hydrogen bond distance are also indicated.

-Figure 5: caption is incomplete

We thank the Reviewer for catching this issue. The figure box was reduced in size upon submission of our manuscript, which partially obscured the figure caption. We have fixed it.

Reviewer #3 (Remarks to the Author):

The manuscript by Maschietto et al. investigates the correlation among the effect of temperature and the presence of an allosteric effector on the dynamics of the Imidazole Glycerol Phosphate Synthase from a thermophilic organism (*T. maritima*).

The work is based on the methodology previously developed and already applied to the same enzyme to unveil the allosteric communication path. The novelty of the present work is the investigation of the effect of temperature. In fact, for this thermophilic enzyme, the allosteric activation is weak at the functional working temperature of the protein as compared to ambient conditions (see Ref. 29 in the manuscript). The authors show that the thermal perturbation on the protein dynamics and flexibility, and network of interactions, is similar to what induced by the effector PRFAR at ambient condition. This explains why the presence of PRFAR at the working condition of the enzyme has a much weaker effect on the enzyme catalytic efficiency.

I find this work interesting since it adds a new piece of information on the relationship among allostery and thermal excitation. The conclusions are adequately supported but extra information (see below) is needed to improve the manuscript and clarify some aspects better.

We thank the Reviewer for the supporting comments on the value and novelty of our work.

I have some points that must be considered before publication.

1. The work focuses only on the allosteric pathway, and the long range communication. However, the boost of the catalytic power should also be considered in terms of reorganization of the active site. The authors should characterize how the presence of the allosteric effector and the higher functional temperature modulate the organization of the catalytic site. How the global reorganization induced by the effector, or the temperature, translates in a higher catalytic efficiency? How was the chemical step of the reaction affected? Ideally, a reader would expect information on the different energetics of the catalytic step for the considered perturbations (PRFAR and temperature).

We really thank the Reviewer for this remark that allowed us realizing that the previous version of the manuscript did not give the deserved attention to the direct implications of our findings on the allosteric mechanism in the active site. We have revised the manuscript to provide the following text (and the figure attached below in the SI), as follows:

Page 11:

The mechanism of Glutamine hydrolysis in the HisH active site involves reorganization of an oxyanion hole^{86,92} to facilitate nucleophilic attack of the cysteine residue (hC84) binding the Gln substrate (e.g. see Fig. 9 in ref. 46). This reorganization in the oxyanion hole region involves nearly 180° rotation of hG50 (of the conserved 49-PGVG sequence) to place a stabilizing HB between the hV51 residue (nearby in sequence) and the Gln substrate, which is possible only upon breaking of the backbone HB existing between hV51(N) and the hP10(O) residue lying in the Ω-loop (see Figure 1). Thus, a strong hV51(N)-hP10(O) HB is typical of the apo state, as it holds the 49-PGVG sequence in an inactive conformation, inapt for catalysis. It was experimentally shown that PRFAR binding causes significant motions in this region of the HisH enzyme (Lipchock and Loria Structure 2010, v.18 1596-1607) and MD simulations showed that such motions are, indeed, the consequence of breaking of the hV51(N)-hP10(O) HB and flipping.⁴⁶

Here, to verify if the temperature increase has effects in the HisH active site similar to those observed for PRFAR binding, we report experimental evidence of chemical shifts alterations induced by temperature (see SI, Figure S12, showing that hG50 resonance move from slow to intermediate, to fast exchange upon temperature increase from 20°C to 50°C) and the time evolution of the critical hV51(N)-hP10(O) HB across the four 1μs MD simulation replicas (see SI, Figure S13). We observed that this critical HB holding the structure in the inactive conformation is essentially broken in apo50, much closer to what found in holo30 (where is often broken but it can be reformed) than to apo30, where is found to be strong, as expected, and only rarely it breaks. These results are, thus, consistent with the presence of a basal activity of IGPS in absence of effector at room temperature (apo30) and they further confirm that the temperature increase to 50°C could affect the catalytic activity by altering the local dynamics in the active site in the same fashion as the effector binding.

2. The authors should compare whether the global structure of the APO enzyme at high temperature gets closer to the structure of the Holo (with PRFAR) enzyme at ambient condition by computing the time evolution of the RMSD of the dynamics of the former with respect to the equilibrated structure of the latter. One should expect a decrease over time. The analysis can be extended to the active site, and also considering structures with the presence of the substrate in the catalytic site (see point 1).

We thank the Reviewer for the suggestion. In fact, we had initially attempted to use the RMSD to show, as suggested by the Reviewer, that the holo30 and the apo50 'active' structures get closer to each other than when comparing these structures to the inactive apo30. However, we found that the RMSD was not a good descriptor for comparing the active and inactive states of IGPS systems. As shown in the Figure below, the RMSD of apo50 with respect to the average structure of holo30 does not necessarily decrease over time in the various replicas. Indeed, this should be expected as shown in previous work (ref. 46 Rivalta et al. PNAS 2012), since the IGPS allosteric communication has a significant component associated with the relative motion between the HisF and HisH proteins (i.e., the so-called 'breathing

motion'). As we demonstrated in this work, the temperature and effector-induced allostery differs exactly in this component (i.e. the dynamical allostery): the temperature increase is not able to control the large fluctuations of the proteins with the same efficacy as the effector does. Thus, the RMSD using the equilibrated structure of holo30 as the reference 'active' structure suffers from a main problem (Figure below): the "equilibrated structure" could not be precisely defined for systems featuring dynamic allostery such IGPS, in the sense that a single, average structure along an equilibrated dynamics cannot represent the "active" state, which is better represented by dynamical fluctuations rather than by a single structure.

3. In the methodology "Eigen Vector Centrality Analysis" it could be useful to stress that the analysis performed on different structures can be directly compared (see results in Fig. 2 Panel a).

We really thank the Reviewer for recognizing this positive aspect of our methodology and for suggesting that we could clarify this in the main text. We have done that, as follows:

Page 4:

Notably, if normalized, the EC coefficients of different states can be used to reliably compare different networks.

4. Binding of PRFAR, pg 7. The authors mention that the Holo structure of the protein is unstable at 50 C and PRFAR unbinds. This must be taken with a grain of salt in view of possible force field artifacts at high temperature. The authors should try to constrain the effector in the binding pocket. Moreover this seems in contradiction with what reported in the Conclusion about the experimental evidence of increased binding affinity of PRFAR at high temperature (if I understand correctly). Also the explanation of the evolutionary gain due to PRFAR increased binding affinity that compensates its weak allosteric activation role at high T is not really clear to me (see Conclusion). If important it must be explained and developed.

This Reviewer's comments parallel points 1) and 4) of Reviewer #1 on PRFAR binding at 50°C and adaptation/compensation mechanism of IGPS.

Regarding the former, we have addressed it by adding a new analysis in the main text and in the (new) Figure SI1 of the SI, as discussed above. We have clarified that MD simulations gave information on the unbinding 'dynamics', showing larger mobility of PRFAR, as expected, with increased temperature. In particular, PRFAR establishes a number of hydrogen bond interactions in *apo50* comparable to those observed in the *holo30* state but more diverse, i.e. with more residues (see Figure SI1 in the SI). It is worth noting that, while we observed experimentally an increase in binding affinity at higher temperature (entropically-driven binding), a complete computational analysis would be required to analyze that observation at the molecular level, including all species competing for binding to the protein, like phosphate ions,

the Gln substrate, the IGP and AICAR product of the complete catalytic reaction, to mention some of them, and their exact concentrations.

Regarding the adaptation/compensation mechanism of IGPS, besides specifying that PRFAR features an entropically-driven binding in the main text, we followed the Reviewer suggestion to clarify the Conclusions and (as reported in the answer to Reviewer #1) we implemented the following changes:

Since it was not mentioned in the main text, we specified the entropically-driven binding of PRFAR, as follows:

Page 2:

The allosteric ligand N'-[(5'-phosphoribulosyl)formimino]-5-aminoimidazole-4-carboxamide-ribonucleotide (PRFAR) features an entropically-driven binding to the HisF subunit, enhancing glutamine hydrolysis at room temperature 5000-fold over its basal level.

Moreover, we modified the Conclusions, as follows:

Page 15:

*Overall, the results presented here provide fundamental insights on the allosteric activation at elevated temperatures, as compared to that induced by PRFAR, which is characterized by less control of protein collective motions yet reminiscent of contacts among critical residues that enable the allosteric signaling transfer. This observation is consistent with previous kinetic data that showed a sizable temperature-dependent increase of the intrinsic enzymatic activity of IGPS, although milder than that induced by the effector.²⁹ In this context, it could be speculated that the entropically-driven PRFAR binding²⁹ represents an evolutionary adaptation strategy to high temperatures by compensating the loss of PRFAR-induced activation with an increased PRFAR binding affinity. However, it is not clear what the physiological needs of *T. maritima* are at its optimal growth temperature vs ambient temperature and thus our speculation should stimulate new studies in this direction. Overall, this study opens the doors for the development of novel tools to control IGPS activity, such as rationally designed allosteric drugs, antipathogens, as well as new engineered variants.*

5. The investigation of the temperature effect on allostery has been investigated also by others in the context of the malate/lactate dehydrogenase protein families. It was shown how temperature mimics the effect of an allosteric effector, and can play a functional regulation role. See: J. Phys. Chem. B (2020) 124, 6, 1001–1008, and Sci. Reports. (2016) 7, 41092.

We thank the Reviewers for suggesting these important references which are relevant to our study and have been added to the Introduction of the revised manuscript.

REVIEWERS' COMMENTS

Reviewer #2 (Remarks to the Author):

all comments have been addressed

Reviewer #3 (Remarks to the Author):

The authors have addressed all the points. The manuscript is ready for publication.